# TMPRSS2 Is Essential for SARS-CoV-2 Beta and Omicron Infection

**DOI:** 10.3390/v15020271

**Published:** 2023-01-18

**Authors:** Kristin Metzdorf, Henning Jacobsen, Marina C. Greweling-Pils, Markus Hoffmann, Tatjana Lüddecke, Felicitas Miller, Lars Melcher, Amy M. Kempf, Inga Nehlmeier, Dunja Bruder, Marek Widera, Sandra Ciesek, Stefan Pöhlmann, Luka Čičin-Šain

**Affiliations:** 1Department of Viral Immunology, Helmholtz Centre for Infection Research, 38124 Braunschweig, Germany; 2Mouse-Pathology, Helmholtz Centre for Infection Research, 38124 Braunschweig, Germany; 3Infection Biology Unit, German Primate Center, 37077 Göttingen, Germany; 4Faculty of Biology and Psychology, Georg-August-University Göttingen, 37073 Göttingen, Germany; 5Immune Regulation Group, Helmholtz Centre for Infection Research, 38124 Braunschweig, Germany; 6Infection Immunology Group, Institute of Medical Microbiology, Infection Prevention and Control, Health Campus Immunology, Infectiology and Inflammation, Otto-Von-Guericke University Magdeburg, 39120 Magdeburg, Germany; 7Institute for Medical Virology, University Hospital, Goethe University Frankfurt, 60596 Frankfurt am Main, Germany; 8DZIF—German Centre for Infection Research, External Partner Site, 60596 Frankfurt, Germany; 9Fraunhofer Institute for Translational Medicine and Pharmacology (ITMP), 60595 Frankfurt, Germany; 10Centre for Individualized Infection Medicine (CIIM), A Joint Venture of Helmholtz Centre for Infection Research and Medical School Hannover, 30625 Hannover, Germany

**Keywords:** SARS-CoV-2, COVID-19, Omicron variant (B.1.1.529), Beta variant (B.1.351), transmembrane serine protease TMPRSS2, mouse, in vivo

## Abstract

The COVID-19 pandemic remains a global health threat and novel antiviral strategies are urgently needed. SARS-CoV-2 employs the cellular serine protease TMPRSS2 for entry into lung cells, and TMPRSS2 inhibitors are being developed for COVID-19 therapy. However, the SARS-CoV-2 Omicron variant, which currently dominates the pandemic, prefers the endo/lysosomal cysteine protease cathepsin L over TMPRSS2 for cell entry, raising doubts as to whether TMPRSS2 inhibitors would be suitable for the treatment of patients infected with the Omicron variant. Nevertheless, the contribution of TMPRSS2 to the spread of SARS-CoV-2 in the infected host is largely unclear. In this study, we show that the loss of TMPRSS2 strongly reduced the replication of the Beta variant in the nose, trachea and lung of C57BL/6 mice, and protected the animals from weight loss and disease. The infection of mice with the Omicron variant did not cause disease, as expected, but again, TMPRSS2 was essential for efficient viral spread in the upper and lower respiratory tract. These results identify the key role of TMPRSS2 in SARS-CoV-2 Beta and Omicron infection, and highlight TMPRSS2 as an attractive target for antiviral intervention.

## 1. Introduction

Severe acute respiratory syndrome coronavirus 2 (SARS-CoV-2), the causative agent of coronavirus disease 2019 (COVID-19), continues to threaten human health. The viral surface protein spike (S) facilitates viral entry into host cells and is the key target of the neutralizing antibody response. For host cell entry, the S protein of SARS-CoV-2 engages the angiotensin-converting enzyme 2 (ACE2) receptor [1,2] and ACE2 expression levels can impact the severity of SARS-CoV-2 infection [3,4]. Further, the entry of SARS-CoV-2 into cells depends on S protein cleavage by a cellular protease, and cell culture data suggest that the cell surface serine protease TMPRSS2 is important for viral entry into lung cells [5,6]. SARS-CoV-2 can also use the endo/lysosomal cysteine protease cathepsin L for S protein activation [7], although the relevance of cathepsin L activity for lung cell entry is controversial.

The constant emergence of SARS-CoV-2 variants that harbor mutations in the S protein, which increase transmissibility and/or antibody escape, is a hallmark of the COVID-19 pandemic. In cell culture, the Alpha, Beta, Gamma and Delta variants depend on TMPRSS2 for lung cell entry [6]. In contrast, the now globally dominating Omicron variant prefers cathepsin L over TMPRSS2, and this preference is associated with reduced lung cell entry [8,9,10]. Further, it has been suggested that inefficient TMPRSS2 usage might account for the inefficient infection of lung cells by the Omicron variant, which, in turn, might explain its attenuation compared to the previously circulating variants [11]. Further, the altered protease choice of the Omicron variant suggests that TMPRSS2 inhibitors, which are currently developed for antiviral therapy, might not be effective against the Omicron variant [12]. However, the role of TMPRSS2 in SARS-CoV-2 spread and pathogenesis in the infected host is largely unclear, considering that inhibitors active against TMPRSS2 usually also block related proteases.

To assess the role of TMPRSS2 in SARS-CoV-2 infection, we used previously described mice that lack TMPRSS2 [13] and have no phenotype in the absence of infection. While the early SARS-CoV-2 variants could only grow in humanized mice expressing the human ACE2 receptor, clinical variants of SARS-CoV-2 with the N501Y mutation in the S protein, including (but not limited to) the Beta variant (B.1.351) and the Omicron variant (B.1.1.529), can infect C57BL/6 mice and transgenic mice with this background (including TMPRSS2^−/−^ mice); this is due to their natural ability to bind to the murine ACE2 receptor [14,15,16]. The infection of C57BL/6 mice with the Beta variant results in disease, while infection with the Omicron variant is asymptomatic [16]. We show that TMPRSS2 is essential for the robust replication of both variants in the upper and lower respiratory tract, although the TMPRSS2-dependence of the Omicron variant was less pronounced in comparison to the Beta variant. Further, only mice expressing TMPRSS2 developed lung inflammation and disease manifestations upon infection with the Beta variant, whereas Omicron infection induced some inflammation, even in TMPRSS2^−/−^ mice. These results indicate the essential role of TMPRSS2 in SARS-CoV-2 Beta and Omicron infection, confirming that the protease is an important target for antiviral intervention.

## 2. Materials and Methods

### 2.1. Animal Experiments

C57BL/6-TMPRSS2^−/−^ [13] and C57BL/6-TMPRSS2^WT/WT^ mice were bred in the animal core facility of the Helmholtz Center for Infection Research, Braunschweig (HZI). C57BL/6J were purchased from commercial vendors (Janvier, Le Genest Saint Isle, France). Animals were housed under Specific Pathogen Free (SPF) conditions at HZI. All performed animal experiments were approved by the Lower Saxony State Office of Consumer Protection and Food Safety, license number: 33.19-42502-04-21/3664.

### 2.2. Cell Culture and Viruses

VeroE6, 293T (human, female, kidney; ACC-635, DSMZ; RRID: CVCL 0063) and BHK-21 cells (Syrian hamster, male, kidney; ATCC Cat# CCL-10; RRID: CVCL_1915, kindly provided by Georg Herrler, University of Veterinary Medicine, Hannover, Germany) were cultured with DMEM (Gibco), supplemented with 5% fetal calf serum (FCS) and 1% penicillin and streptomycin (P/S). The SARS-CoV variants, Beta (GenBank accession number: MW822592 (B.1.351; FFM-ZAF1/2021) [17] and Omicron BA.1 (GISAID: EPI_ISL_6959871, FFM-SIM/0550/2021) [18], were propagated using CaCo-2 cells as described previously [18]. The integrity of the viral genomes was confirmed by Nanopore sequencing. Cell culture supernatants were used for infection experiments after removing cellular debris by centrifugation. Titrations of viral stocks were performed on VeroE6 cells cultured in DMEM, supplemented with 5% FCS and 1% P/S.

### 2.3. Expression Plasmids

Plasmids pCAGGS-VSV-G (vesicular stomatitis virus glycoprotein) [19], pQCXIP-mACE2-cMYC [20], pcDNA3.1-mTMPRSS2 [21], pCG1- SARS-CoV-2 B.1 SΔ18 (codon-optimized, C-terminal truncation of 18 amino acid residues, GISAID Accession ID: EPI_ISL_425259) [22], pCG1-SARS-CoV-2 B.1.351 SΔ18 (GISAID Accession ID: EPI_ISL_700428) [23] and pCG1-SARS-CoV-2 BA.1 SΔ18 (GISAID Accession ID: EPI_ISL_6640919) [20] have been described previously.

### 2.4. SARS-CoV-2 Infection Experiments and Organ Processing

In this study, 16–18-month-old female and male mice were intraperitoneally (i.p.) anesthetized with a mixture of Ketamine (80 mg/g body mass) and Xylazine (10 mg/g body mass) in a 0.9% NaCl solution. Under deep anesthesia, mice were intranasally (i.n.) infected with 10^4^ plaque forming units (PFU) of the respective virus, diluted in a total volume of 20 µL phosphate buffered saline (PBS). Control animals received 20 µL PBS. The body mass and clinical condition were monitored for three days following infection. The behavior of the mice was evaluated according to five parameters, including provoked and spontaneous behavior, posture and fur coat. In addition, any change in body mass was assessed, which was tolerable up to 20% mass loss. The parameters were quantified from 0 (no abnormality) to 3 (humane endpoint) and were accumulated to a cumulative score.

Three days post infection (dpi), animals were euthanized by CO_2_ inhalation, blood was collected from the heart and nasal washes were performed with 200 µL sterile PBS. The lung, trachea and brain were isolated for further analyses. Strict sterile techniques were applied to avoid the carryover of the virus from one organ to another and between animals. Right lungs were isolated for homogenization, while left lungs were isolated in 4% formalin solution (in PBS) for histology. Blood clotting occurred at room temperature (RT) for 30–60 min and a serum was isolated by centrifugation. Nasal washes were purified from debris by centrifugation. Solid organs were weighed and homogenized in bead-containing lysis tubes (Lysis tube E, Analytik Jena) filled with 500 µL (trachea) or 1000 µL (lung and brain) of sterile PBS in a MP Biomedical FastPrep 24 Tissue Homogenizer (MP Biomedicals, CA, USA) (full speed, 2 × 20 s). After centrifugation, supernatants were aliquoted for further analyses. Samples designated for RNA-isolation were mixed with Trizol reagent (Invitrogen) at a 1:3 ratio. All samples were stored at −80 °C until use without multiple freeze-thaw cycles. All centrifugation steps were carried out at 10,000× *g* for 10 min at 4 °C.

### 2.5. Determination of SARS-CoV-2 Viral Load and IFNs by qRT-PCR

RNA isolation was performed with the innuPrep Virus TS RNA kit (Analytik Jena), according to the manufacturer’s protocol. Briefly, Trizol-inactivated samples were mixed with an equal volume of lysis solution *CBV*, containing carrier RNA and Proteinase K, and incubated at 70 °C for 10 min. After lysis, samples were mixed with two sample volumes of isopropanol. The sample was then applied to the provided spin filters and washed with washing buffer (LS) and 80% ethanol. RNA was eluted in 60 µL RNase-free water and stored at −80 °C after the RNA concentration was determined with a NanoDrop (Thermo Scientific NanoDropOne). 

SARS-CoV-2 N-gene copy numbers were determined by qRT-PCR, using the following kits: 2019-nCoV RUO primer kit (IDT), rodent GAPDH Taq man (Applied Biosystems), Taq Path 1 step RT-qPCR Master (Applied Biosystems) and CoV RUO positive control kit (IDT), following the manufacturer’s protocols. SARS-CoV-2 N-gene copy numbers were normalized to total RNA input and to rodent GAPDH copy numbers.

The mRNA expression of RSP9, IFNβ and IFNα2 were detected by using the SensiFAST™ SYBR^®^ No-ROX One-Step Kit (meridian Bioscience), following the manufacturer’s protocol on 3-step cycling. IFN RNA was quantified using the IFNα2 forward 5′-AGAGCCTTGACACTCCTGGTA-3′ and reverse 5′-CTCAGGACAGGGATGGCTTG-3′ primer, and INFβ forward 5′-AACTCCACCAGCAGACAGTG-3′ and reverse 5′-GGTACCTTTGCACCCTCCAG-3′ primer, respectively. Gene expression was analyzed via the ∆∆Ct-method and normalized to the endogenous RSP9 expression of the PBS control group, respectively. 

### 2.6. Determination of SARS-CoV-2 Live-Virus Titers by Plaque Assay

Organ homogenates and nasal washes were serially diluted in infection media (DMEM, 5% FCS, 1% P/S) in a 96-well format. Sample dilutions were transferred onto confluent VeroE6 cells, seeded the day before, in infection media. After inoculation at 37 °C for 1 h, the supernatant was removed and 1.5% methylcellulose in MEM media, supplemented with 5% FCS, 1% L-Glutamine and 1% P/S (final concentration), was added to the cells. Lung and brain tissue homogenates were tested, starting with undiluted material and nasal washes, with a 1:10 dilution for maximum assay sensitivity.

Cells were incubated at 37 °C for 48 h before the supernatant was removed and plates were inactivated in a 4% formalin solution in PBS for 10 min. Fixed cells were subjected to immunofluorescence staining against the SARS-CoV-2 N protein. Briefly, fixed cells were blocked at RT with 1% BSA (Sigma) in PBS for 30 min and permeabilized with 0.1% Triton-X100 (Sigma) at RT for 30 min. Cells were incubated with a rabbit anti-SARS-CoV-2 nucleocapsid antibody (Abcalis, ABK84-E03-M; 0.23 mg/mL) at RT for 30 min. After washing the cells three times with PBS-T (PBS, 0.05% Tween-20), the secondary antibody anti-rabbit Fab Alexa488 (Cell Signaling Technology) was added for 30 min at RT. After washing the cells three times with PBS-T, stained cells were visualized using an IncuCyte (Sartorius; GUI software versions 2019B Rev1 and 2021B). 

### 2.7. Histology

Approx. 3-µm-thick sections of formalin-fixed, paraffin-embedded (FFPE) left lung tissue were stained with hematoxylin-eosin (HE), according to standard laboratory procedures. The HE stained sections were evaluated, blinded and randomized by a trained veterinarian using a score and the area affected by the pathological change was estimated as follows: 1 = up to 30%, 2 = 40–70%, 3 = more than 70%. The severity of the parameters, alveolar edema, interstitial pneumonia, broncho-alveolar inflammation, perivascular inflammation, vasculitis, were graded as follows: 1 = mild, 2 = moderate and 3 = severe. The score for interstitial pneumonia, broncho-alveolar inflammation, perivascular inflammation and vasculitis were summarized to an inflammation score of 0–15. The presence of inflammatory cells was estimated for lymphocytes, neutrophils and macrophages: 1 = occasionally seen, 2 = easily visible, large amounts, and 3 = dominating inflammatory cell. The scores for lymphocytes, neutrophils and macrophages are qualitative markers and can, therefore, not be summarized in one score. Analysis was performed randomized and blinded. 

### 2.8. Immunofluorescence against SARS-CoV-2 Nucleocapsid and MAC2

For 2plex immune-fluorescence staining, 3-µm-thick FFPE sections were stained for SARS-CoV-2 with mouse-anti-nucleocapsid CoV-1/2 (Synaptic Systems, HS-452 11, clone 53E2, subtype: IgG2a) and for macrophages with rat-anti-mouse-MAC2 (Biozol Diagnostica/CEDARLANE, CL8942AP, clone M3/38). The slides were scanned with an Olympus VSI120 whole slide scanner using the Software VS-ASW 2.9.2 (Built 17,565). Scans were achieved with a 20× via (Maximum Intensity Projection) Z mode with 3 layers, and automatically analyzed with QuPath 0.32 (The University of Edinburgh) [24]. For analysis, the regions of interest (ROIs) were determined automatically by simple tissue detection. The threshold for the detection of positive cells was set to 2-fold background, with a minimum area of 10,000 µm^2^. Purely connective tissue areas, fat, intrapulmonary bronchi (due to auto-fluorescence in Texas red of epithelium), and areas that showed large hemorrhages, were excluded. Subsequently, cells were classified and counted using an object classifier for Texas Red (MACc2) and for FITC (SARS-CoV-2). The results were exported and processed in Excel to calculate the percentage of SARS-CoV-2-stained cells per total cells, or per MAC2 positive cells.

### 2.9. VSV Pseudotyped Particles 

Vesicular stomatitis virus (VSV) pseudotype particles bearing SARS-CoV-2 S proteins were produced according to an established protocol [25]. In brief, 293T cells expressing SARS-CoV-2 S protein, VSV-G (vesicular stomatitis virus glycoprotein) or no viral surface protein (negative control) following transfection were inoculated with VSV-G-transcomplemented, firefly luciferase (FLuc)-encoding VSVΔ*G(FLuc) [26] (kindly provided by Gert Zimmer, Institute of Virology and Immunology, Mittelhäusern, Switzerland) and incubated for 1 h at 37 °C and with 5% CO_2_. Next, cells were washed with phosphate-buffered saline (PBS) and a medium containing the anti-VSV-G antibody (culture supernatant from I1-hybridoma cells; ATCC no. CRL-2700; 1:1000) was added to all cells, except for those expressing VSV-G (for these cells medium without antibody was added). Following an incubation period of 16–18 h, supernatants were collected, cleared from cellular debris by centrifugation (4000× *g*, 10 min, room temperature), and stored at −80 °C until further use.

For transduction experiments, BHK-21 cells were seeded in 96-well plates. At 24 h before transduction, cells were transfected with an empty expression plasmid, mACE2 expression plasmid, mTMPRSS2 expression plasmid, or a combination of mACE2 and mTMPRSS2 expression plasmids, using Lipofectamine 2000 (Thermo Fisher Scientific). Immediately before transduction, cells were washed with PBS and incubated with regular culture medium or a culture medium containing 50 mM ammonium chloride (Sigma-Aldrich). After an incubation period of 1 h, pseudotype particles were added to the cells and incubated for 16–18 h at 37 °C and with 5% CO_2_, before transduction efficiency was analyzed by measuring FLuc activity in cell lysates. Cells were lysed by incubation with PBS, containing 0.5% Triton-X100 (Carl Roth), for 30 min at room temperature. Next, cell lysates were transferred into white 96-well plates and FLuc substrate (Beetle-Juice, PJK) was added. Luminescence was recorded using a Hidex Sense plate luminometer (Hidex).

### 2.10. Statistics

Data were represented as mean ± SEM, mean and mean ± SD, and statistically analyzed using the GraphPad Prism 9.0. Statistics for score and a two-way ANOVA with Geisser-Greenhouse correction for body mass; then, the Tukey multiple comparison test was applied. For further analysis comparing two groups, the statistical significance was calculated by a two-tailed Welch’s test.

## 3. Results

### 3.1. TMPRSS2 Is Essential for Pathogenesis Upon Infection with the SARS-CoV-2 Beta Variant

To investigate the role of the serine protease TMPRSS2 after infection with SARS-CoV-2, C57BL/6 wild-type (wt) and TMPRSS2 knock-out (TMPRSS2 KO) mice were infected intranasally (i.n.) with SARS-CoV-2 Beta (B.1.351) or Omicron (B.1.1.529) and monitored for 3 days (Figure 1A); during this time, the mass loss and clinical score were determined (Figure 1B–I). We chose the Beta variant, rather than the Alpha or Gamma, because it grows better than those variants in non-transgenic mice [27]. Body mass was reduced by 5% at 2 days and 10% at 3 days post infection (dpi) with the Beta variant (Figure 1B). In contrast, infected TMPRSS2 KO mice and control mice, receiving PBS instead of virus, showed no mass loss (Figure 1B,C). Further, wt mice infected with the Beta variant had significantly higher clinical scores than the infected TMPRSS2 KO animals at 2 and 3 dpi, these mice exhibited clinical scores largely identical to those of the PBS-treated controls (Figure 1D,E). In contrast, we observed no differences in mass loss (Figure 1F,G) or in clinical scores (Figure 1H,I) in the Omicron-infected mice, compared to the PBS control group; this is in line with other studies. Nor did we observe differences between the wt and TMPRSS2 KO mice, independent of the animal age (Appendix A). In sum, our results indicated that expression of the serine protease TMPRSS2 was critical for disease development, upon infection with the Beta variant SARS-CoV-2.

### 3.2. TMPRSS2 Is Essential for Robust Spread of the SARS-CoV-2 Omicron and, Particularly, Beta Variant

The viral load in the lower respiratory tract correlates with the progression of the disease and severity of the symptoms [28,29]. Hence, we measured the viral loads in the lungs, trachea and nasal washes at 3 dpi. In wt mice infected with the SARS-CoV-2 Beta variant, SARS-CoV-2 RNA was readily detectable in the nose, trachea and lung (Figure 2A), while in infected TMPRSS2 KO mice, RNA levels were roughly 100- (nose) to 100,000-fold (lungs) reduced or undetectable (trachea). Further, the viral RNA load in the trachea was 100–1000-fold lower than in the lungs, and a similar trend was observed when the infectious viral load was determined (Figure 2C); this indicates that the virus proliferated more efficiently in the lungs than in the upper respiratory tract in our experimental model system. In sum, these results, and the results discussed above (Figure 1), show that TMPRSS2 is essential for the robust spread and pathogenesis of the SARS-CoV-2 Beta variant.

The relative viral loads, detected upon infection of wt mice with the SARS-CoV-2 Omicron variant, mirrored the findings for the Beta variant; however, the overall virus load was roughly 10 to 100-fold reduced in comparison to the Beta variant (Figure 2B). Interestingly, the viral RNA loads were markedly lower in TMPRSS2 KO mice compared to wt mice, although the effects were not as pronounced as those observed for the SARS-CoV-2 Beta variant (Figure 2A,B). Thus, the viral RNA load was reduced at least 100-fold in the lungs of TMPRSS2 KO, relative to wt mice, and was close to background levels in the trachea and nose (Figure 2A,B). Interestingly, no SARS-CoV-2 was detectable in the brains of infected mice, either as RNA or infectious virus; this was also observed with the Beta and Omicron infections (Appendix A). Finally, no infectious Omicron virus was detected at all, even among the wt mice (Figure 2D). These results indicate that TMPRSS2 is also essential for the robust spread of the SARS-CoV-2 Omicron variant, although the TMPRSS2-dependence of the Omicron variant is not as pronounced as it is for the SARS-CoV-2 Beta variant.

### 3.3. TMPRSS2 Promotes Lung Cell Infection by the SARS-CoV-2 Beta and Omicron Variants

To visualize the presence of SARS-CoV-2 in lungs, lung samples were stained for nucleocapsid of SARS-CoV-2 (green). In addition, macrophages (magenta) were stained with MAC2. In the lungs of wt mice infected with SARS-CoV-2 Beta variant, the virus was readily detected, particularly in the bronchi (Figure 3A). Quantification revealed an increased proportion of lung cells that contained the SARS-CoV-2 nucleocapsid, in comparison to the total cells counted (Figure 3B). Furthermore, the viral N protein was detected in macrophages after infection of wt mice with the Beta variant (Figure 3C). The number of SARS-CoV-2 Beta-infected lung cells was much lower in TMPRSS2 KO mice, underlining the essential role of TMPRSS2 in infection with the SARS-CoV-2 Beta variant. The SARS-CoV-2 Omicron variant was detected in lung cells of wt mice as well, albeit at a lower level than the Beta variant, while no virus was detected in samples from TMPRSS2 KO mice (Figure 3B). Finally, the SARS-CoV-2 Omicron variant was also detected in macrophages (Figure 3C). These results confirm the TMPRSS2-dependent lung cell infection of the SARS-CoV-2 Beta variant, and further support our finding that TMPRSS2 is also essential for the robust lung cell infection of the SARS-CoV-2 Omicron variant.

### 3.4. TMPRSS2 Plays a Crucial Role in Inflammatory Lung Damages of SARS-CoV-2 Beta Infected Mice

In order to assess lung pathology after SARS-CoV-2 infection, histopathological sections of the lungs were stained with haematoxylin-eosin (HE) and scored for inflammatory alterations of the pulmonary structure. Prominent inflammatory infiltrates were seen in the Beta-infected wt mice (Figure 4A,B), which were substantially reduced in TMPRSS2 KO animals. The inflammatory lesions were minor in Omicron-infected animals, irrespective of TMPRSS2 presence (Figure 4A,B). Further, infection with the SARS-CoV-2 Beta variant lead to a marked infiltration of lymphocytes, neutrophils and macrophages into the lungs of wt, but not TMPRSS2 KO mice; meanwhile, infection-mediated infiltration, measured for wt and TMPRSS2 KO mice, were within the background range (Figure 4A,C–E). Finally, type I interferon responses were detected in the lungs of Beta- and Omicron-infected wt but not TMPRSS2 KO- or PBS-treated wt mice; this further indicates that the Beta variant in TMPRSS2 KO mice simply does not disseminate into the lung, which is very sensitively indicated by the absence of a type I IFN response (Appendix A), as measured by qPCR for IFN transcripts. These results indicate that TMPRSS2 is essential for lung pathogenesis upon infection with the SARS-CoV-2 Beta variant.

### 3.5. Omicron Spike-Driven Cell Entry Is Less Dependent on TMPRSS2 Than Beta

To experimentally inhibit the endosomal pathway, we used ammonium chloride (NH_4_Cl), elevating the endosomal pH and thereby preventing the proteolytic cleavage of SARS-CoV-2 (Figure 5A). The VSV glycoprotein (VSV-G) showed a very efficient cell entry, independent of the presence of murine (m)ACE2 and (m)TMPRSS2 in the DMSO control (grey bars); meanwhile, the cell entry was severely limited when the endosomal route was inhibited by NH_4_Cl (green bars) (Figure 5B). The expression of mACE2 and/or mTMPRSS2 did not facilitate the entry of VSV expressing B.1 spikes, and accordingly, the inhibition by NH_4_Cl had no effect on the pseudotype entry (Figure 5C). The cell entry of Beta VSV pseudotypes efficiently rescued viral entry, driven by the S protein of the Beta variant, from inhibition by NH_4_Cl (Figure 5D). Notably, the NH_4_Cl-mediated reduction was only partially rescued by the expression of TMPRSS2 in Omicron VSV pseudotypes (Figure 5E). Therefore, the mTMPRSS2 rescue efficiency was higher in the Beta than in the Omicron variant (Figure 5F). In conclusion, both Beta and Omicron efficiently infected cells using the TMPRSS2-dependent and the TMPRSS2-negative pathway, but TMPRSS2 was sufficient for Beta spike-driven entry, whereas Omicron was only partially rescued.

## 4. Discussion

This study examined the role of TMPRSS2 in SARS-CoV-2 infection by using genetic knock-out mice. The approach, unlike studies with protease inhibitors, allowed this study to reveal the contribution of TMPRSS2 to SARS-CoV-2 infection, without interfering with the other proteolytic enzymes of the host cell. Our results show that TMPRSS2 is essential for the spread and pathogenesis of the SARS-CoV-2 Beta variant in the respiratory tract. Further, we found that, despite a preference for cathepsin L in cell culture models, the SARS-CoV-2 Omicron variant depends on TMPRSS2 for spread in the respiratory tract, although the dependence is not as pronounced as for the Beta variant. Thus, TMPRSS2 is an essential host cell factor in the context of SARS-CoV-2 infection and constitutes an attractive target for antiviral intervention.

Our results agree with the recent publication of Iwata-Yoshikawa et al. [30], who demonstrated the critical role of TMPRSS2 in Beta and Omicron’s in vivo replication. Nevertheless, while Iwata-Yoshikawa et al. claim that Omicron is no exception to using TMPRSS2 for murine airway infection, we performed a head to head comparison of Beta and Omicron sub-genomic RNA loads in respiratory organs; we observed that Omicron was less dependent on TMPRSS2 in vivo than the Beta variant. Interestingly, Iwata-Yoshikawa et al. reported a similar dependence of Beta and Omicron on endosomal entry in vitro, whereas we observed that the endosomal entry pathway was less relevant for cell-entry for the Beta than of the Omicron variant in vitro; this was in accordance with other publications [8,9,10] and our in vivo data. Finally, since it was reported that TMPRSS2 expression levels vary according to age, we went beyond the previous paper [30] by comparing adult and old mice lacking TMPRSS2. Somewhat surprisingly, we observed a similar dependence of both tested variants in TMPRSS2 in young and in old mice. Therefore, while both studies demonstrate the role of TMPRSS2 in the in vivo growth of SARS-CoV-2 variants, our study adds important novel detail to our understanding of these cellular and molecular processes. 

We employed wt mice for analysis of the role of TMPRSS2 in SARS-CoV-2 infection. Infection of these animals with SARS-CoV-2 does not depend on directed ACE2 expression and thus avoids phenomena linked to receptor overexpression, including virus infection of neuronal tissue; this is detected with K18-ACE2 mice, but not observed in C57BL/6 mice in our study, at least not at dpi 3. Wt mice are permissive to infection with the Beta and Omicron variants, which harbor mutations in their S proteins [15,27] that allow for the usage of murine ACE2 (Figure 5). Thus, murine ACE2 and TMPRSS2 reflect important aspects of cell entry, supported by human ACE2 and TMPRSS2, indicating that the mouse model used here should be suitable for mirroring SARS-CoV-2’s entry into the human respiratory epithelium.

The essential role of TMPRSS2 in SARS-CoV-2’s entry into lung cells was first described with cell lines [2,7,31], and subsequent studies confirmed these results with organoids [32] or primary respiratory epithelial cell cultures [33]. However, when two of the protease inhibitors, Camostat and Nafamostat, employed to demonstrate an important role of TMPRSS2 in SARS-CoV-2 infection in cell culture [5], were employed for COVID-19 treatment in humans, moderate or no efficacy was observed [34]. While the failure of these compounds to appreciably inhibit SARS-CoV-2 spread in humans might be for several reasons, including a late treatment start and insufficient inhibitor concentrations in the respiratory tract, they raised the question of whether TMPRSS2 is, indeed, essential for viral spread in the host and is thus a suitable therapeutic target. Our findings indicate is the following: TMPRSS2 was essential for the spread and pathogenesis of the SARS-CoV-2 Beta variant in the murine respiratory tract. These findings are in line with that of a recent study and with the finding that inhibitors with a higher potency and specificity than Camostat, are protective against the spread and pathogenesis of SARS-CoV-2 in mice [35]. We expect that our findings can be extrapolated to the Alpha, Gamma and Delta variants, which all fully depend on TMPRSS2 for lung cell infection in cell culture.

The Omicron variant is the first variant of concern that prefers cathepsin L over TMPRSS2 for entry into cell lines [10]. Interestingly, this altered protease choice is associated with reduced lung cell entry and it has been speculated that the inability to efficiently use TMPRSS2 might account for the inefficient lung cell infection of the Omicron variant; this might, in turn, be partially responsible for the attenuation of the Omicron variant, relative to the previous circulating variants of concern [7,9,10,36].

Our study has some limitations. We focused on the BA.1 Omicron subvariant and did not analyze the presently circulating Omicron subvariants, some of which show increased lung cell entry. Moreover, it is possible that additional proteases in humans (but not in mice) may act in a redundant manner to TMPRSS2. In that case, TMPRSS2 would be essential for virus in vivo growth only in a murine system.

In conclusion, our study reveals the highly important role of TMPRSS2 for the replication of clinically relevant SARS-CoV-2 variants, and a hierarchy of relevance that is reflected by the ability of the variants to utilize the endosomal route as an alternative entry mechanism. Therefore, this study adds important insights to the knowledge regarding the progression and mechanism of SARS-CoV-2 disease.

## Figures and Tables

**Figure 1 viruses-15-00271-f001:**
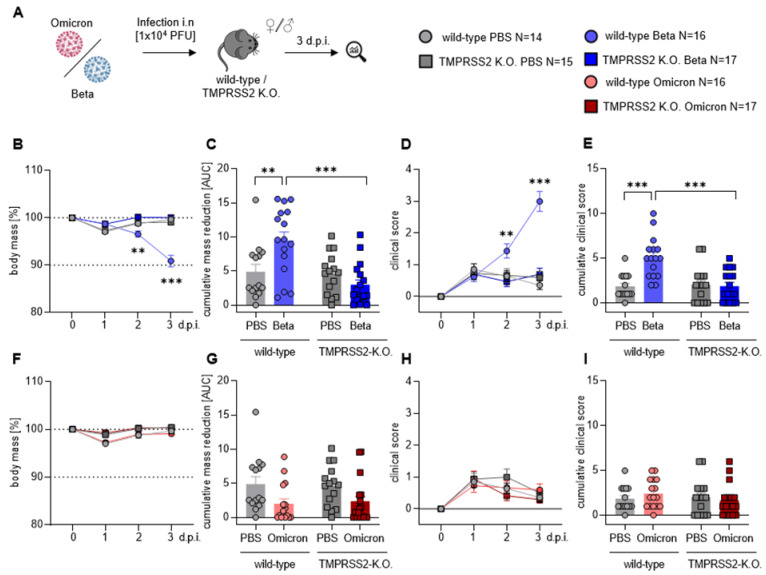
TMPRSS2 is essential for pathogenesis upon infection with the SARS-CoV-2 Beta variant. (**A**) Wt and TMPRSS2 KO mice were i.n. infected with 10^4^ PFU of SARS-CoV-2 Beta or Omicron. Uninfected control mice were i.n. treated with PBS. Body mass and clinical scores were monitored until 3 dpi (**B**,**F**) Relative body mass of SARS-CoV-2 Beta (**B**) or Omicron (**F**) infected mice. (**C**,**G**) Cumulative relative mass reduction in SARS-CoV-2 Beta (**C**) or Omicron (**G**) infected mice until 3 dpi are shown as area under the curve (AUC). Data for individual mice are shown as symbols. (**D**,**H**) Clinical scores of SARS-CoV-2 Beta (**D**) or Omicron (**H**)-infected mice. (**E**,**I**) Cumulative clinical score in SARS-CoV-2 Beta (**E**) or Omicron (**I**) infected mice until 3 dpi. Data for individual mice are shown as symbols. (**B**–**I**) Data are plotted as means and error bars denote SEM. Statistical significance for sequential body mass and clinical scores (**A**,**D**,**F**,**H**) was calculated using two-way ANOVA with Geisser–Greenhouse correction, and the correction for multiple comparisons was calculated following Tukey. Statistical significance for cumulative data sets (**C**,**E**,**G**,**I**) was calculated with Welch’s *t* test (two-tailed) (** *p* < 0.01, *** *p* < 0.001).

**Figure 2 viruses-15-00271-f002:**
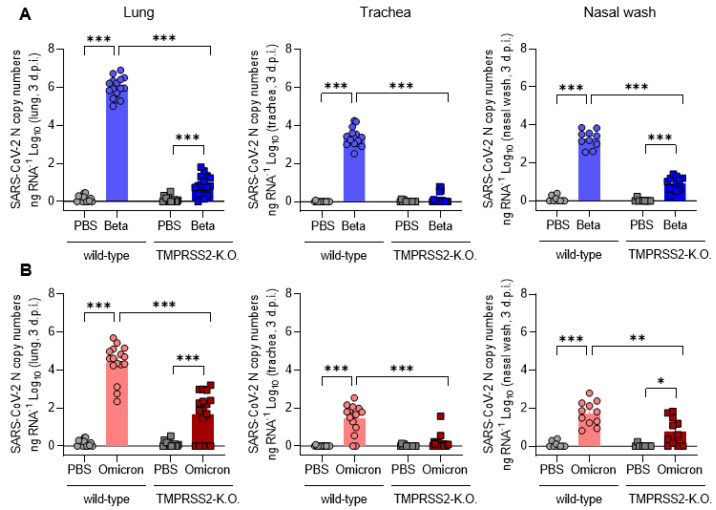
TMPRSS2 is essential for the robust spread of the SARS-CoV-2 Omicron and, particularly, the Beta variant. (**A**) SARS-CoV-2 N gene copy numbers as copy number/ng RNA Log10 at 3 dpi with SARS-CoV-2 Beta in lung homogenates (left), trachea homogenates (middle) and nasal washes (right). (**B**) SARS-CoV-2 N gene copy numbers as copy number/ng RNA Log10 at 3 dpi with SARS-CoV-2 Omicron in lung homogenates (left), trachea homogenates (middle) and nasal washes (right). (**C**) SARS-CoV-2 Beta infectious-virus titer as PFU/mL Log10 at 3 dpi in lung homogenates (left), trachea homogenates (middle) and nasal washes (right). (**D**) SARS-CoV-2 Omicron infectious-virus titer as PFU/mL Log10 at 3 dpi in lung homogenates (left), trachea homogenates (middle) and nasal washes (right). Dotted lines indicate the limit of detection (LOD) for live-virus titers (**C**,**D**). Statistical significance for was calculated using Welch’s *t* test (two-tailed) (* *p* < 0.05, ** *p* <0.01, *** *p* < 0.001).

**Figure 3 viruses-15-00271-f003:**
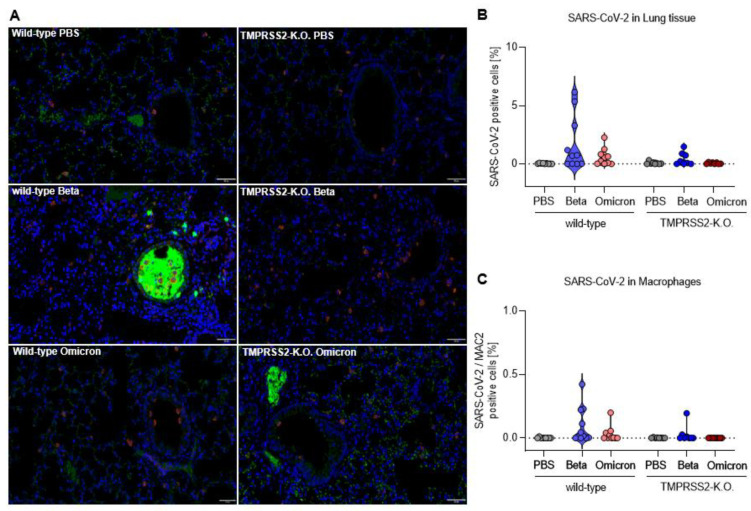
TMPRSS2 promotes lung cell infection by the SARS-CoV-2 Beta and Omicron variants. (**A**) Representative images of immunohistochemical staining for the Nucleocapsid CoV-1/2 (green) and MAC2 (magenta) in lung tissue of PBS (top), Beta (middle) and Omicron (bottom) infected wilt-type (left) and TMPRSS2 KO mice (right). All of the images were equally increased in brightness and contrast by the same absolute values. Scale bars, 50 µm. (**B**) Number of SARS-CoV-2 positive cells as a percentage of total cells counted. (**C**) Number of macrophages immunohistochemically stained for MAC2 that are concurrently positive for SARS-CoV-2 as a percentage.

**Figure 4 viruses-15-00271-f004:**
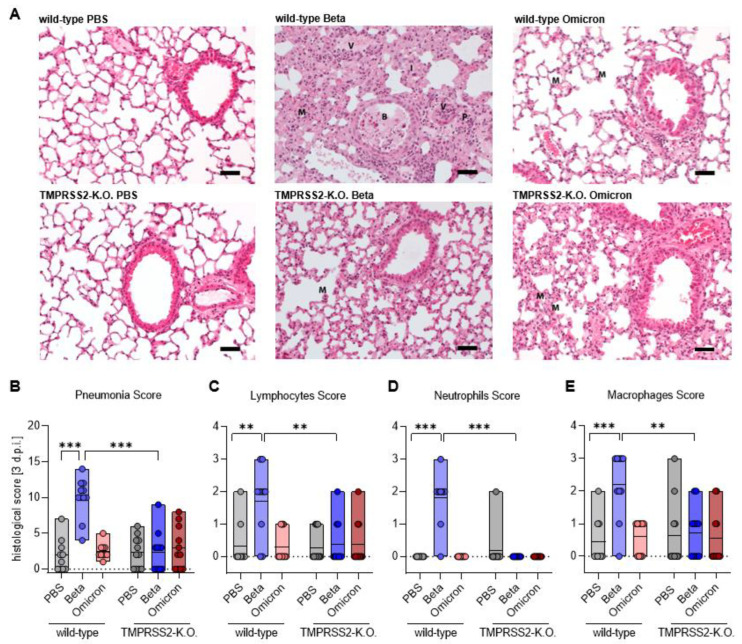
TMPRSS2 plays a crucial role in inflammatory lung damages of SARS-CoV-2 Beta infected mice. (**A**) Representative hematoxylin-eosin (HE) staining of lung tissue from PBS (top), Beta (middle) and Omicron (bottom)-infected wild-type (left) and TMPRSS2 KO mice (right). I = interstitial pneumonia, B = broncho-alveolar inflammation, P = perivascular inflammation and V = vasculitis, M = macrophages representing the severity of the individual parameters summarized in the Pneumonia score. Scale bar 50 µm. (**B**) For the pneumonia score, the following scores were cumulated: area affected by the pathological change, interstitial pneumonia, broncho-alveolar inflammation, perivascular inflammation and vasculitis. The severity of the parameters was classified as: 1 = mild, 2 = moderate 3 = severe. (**C**–**E**) The presence of inflammatory cells was estimated for: (**C**) Lymphocytes (left), (**D**) neutrophils (middle) and (**E**) macrophages (right): 1 = occasionally seen, 2 = easily visible, large amounts 3 = dominating inflammatory cell. (**B**–**E**) Data are shown as median. Statistical significance for was calculated using Welch’s *t* test (two-tailed) (** *p* < 0.01, *** *p* < 0.001).

**Figure 5 viruses-15-00271-f005:**
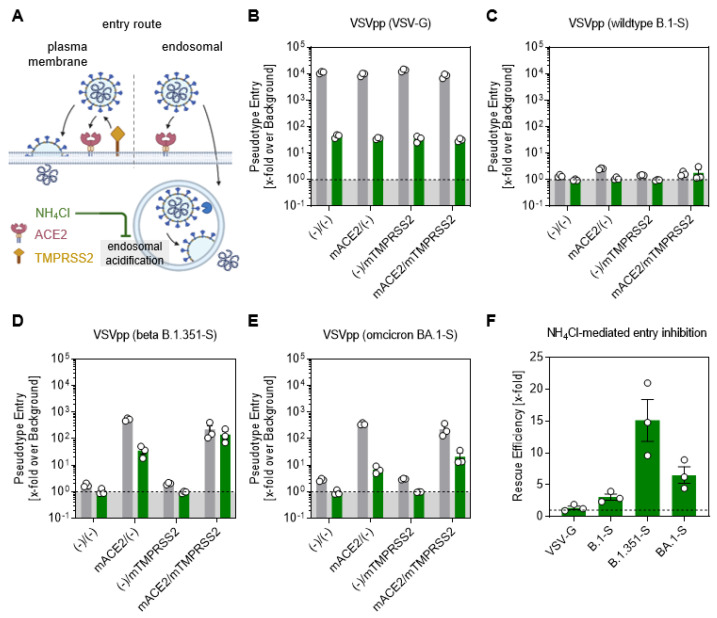
Omicron spike-driven cell entry is less dependent on TMPRSS2 than Beta. (**A**) Simplified representation of the two putative entry routes for SARS-CoV-2 via plasma membrane fusion or via the endosomal pathway. In both routes, SARS-CoV-2 is recognized by ACE2 and either proteolytically cleaved in the presence of TMPRSS2 to enter the cell via membrane fusion. In the absence of TMPRSS2, the virus is engulfed endosomally and, after proteolytic cleavage by cathepsin L, membrane fusion of the host endosome with the viral membrane is induced. Ammonium chloride (NH_4_Cl) prevents endosomal acidification, making the endosomal entry pathway inefficient for the virus entry. BHK21 cells transiently expressing murine (m)ACE2 and/or (m)TMPRSS2 were treated for 2 h with DMSO as control (grey bar) or 50 mM NH_4_Cl (green bar). Furthermore, they were inoculated with particles pseudotyped with (**B**) the glycoprotein, (**C**) SARS-2-S of wild-type B.1 (**D**) Beta B 1.351-S and (**E**) Omicron BA.1-S. At 16 h after inoculation, the SARS-2-S-related cell entry of the viral pseudotypes was analyzed by measuring the activity of luciferase activity encoded by the virus in cell lysates. (**F**) The calculated rescue efficiency represented by the fold difference in NH_4_Cl-mediated entry inhibition between mACE2/(−) and mACE2/mTMPRSS2 groups. (Figure 5A Created with BioRender.com).

## Data Availability

All relevant data are deposited in the manuscript and Appendix A.

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
