# Peer review of "TMPRSS2 Is Essential for SARS-CoV-2 Beta and Omicron Infection"

_viruses, 2023, doi:10.3390/v15020271_

Round 1

Reviewer 1 Report

If possible, insert reference numbers at the end of sentences.

Author Response

Reviewer 1)

17 Dec 2022 17:25:31

Reviewer 1 was extremely positive about our manuscript and had not objections, save the one listed below:

If possible, insert reference numbers at the end of sentences.

We moved the references to the end of the sentences, where possible, but we think it is useful to integrate the references into the text when multiple references relate to different statements within a sentence, so that the reader does not have to go through all the references to find the requested information.

Reviewer 2 Report

This manuscript by Kristin Metzdorf, Luka Čičin-Šaine et al. examined the role of TMPRSS2 in SARS-CoV-2 infection by using genetic knock-out mice without the protease. They focused on the role of the protease on the Beta (B.1.351) and the Omicron variant (B.1.1.529)  and found a moderately different effect on the two variants. Based on a series of studies (cell culture, virus load, animals,et al.) and analysis, their data showed a critical role of TMPRSS2 in SARS-CoV-2 Beta and Omicron infection, and they proposed that the protease can still be an important target for antiviral intervention for the two variants. Furthermore, this approach using knock-out mouse, unlike studies with protease inhibitors, reveals the contribution of TMPRSS2 to SARS-CoV-2 infection without interfering with other proteolytic enzymes of the host cell.  In particular, they found, despite a preference for cathepsin L in cell culture models, the SARS-CoV-2 Omicron variant depends on TMPRSS2 for the spread in the respiratory tract, although the dependence is not as pronounced as for the Beta variant. 

Overall, I think their experiment is well designed and these authors considered many aspects of the effect and performed corresponding measurements. Moreover, considering the importance of SARS-CoV-2, especially its rapid mutation, knowledge of these variants is critical. And I believe the public as well as the academics need to know all the experiments and results asap. In fact, the role of TMPRSS2 in SARS-CoV-2 Omicron infection and its inhibitors for antiviral strategy is of great interest, as these authors claimed. Thus,  I support the publication of this work after they answered my following concerns. 

1. It is no doubt that Omicron should be closed to test the TMPRSS2's effect. However, I cannot find why these authors chose the Beta variant, especially at the beginning.  There is indeed one sentence for this part at the beginning of the manuscript.  "While the early SARS-CoV-2 variants could only grow in humanized mice expressing the human ACE2 receptor, the Beta variant (B.1.351) and the Omicron variant (B.1.1.529) can infect C57BL/6 mice and trangenic mice on this background (including TMPRSS2-/- mice) due to naturally occurring S protein mutation N501Y within the receptor binding domain (RBD) ." However, alpha and gamma variants contain the N501Y mutation on their RBD (1-2). So, the authors should elaborate on the reasoning. 

1. N501Y mutation of spike protein in sars-cov-2 strengthens its binding to receptor ace2. Elife 2021, 10, e69091.

 2. The N501Y spike substitution enhances sars-cov-2 infection and transmission. Nature 2022, 602, 294-299.

2. In this work, the authors examined the effect of TMPRSS2 using mouse model. Although many effect of the protease for Omicron is still observed in mouse, the effect on humans may be slightly different. This point should be kept in mind.    

3. Page 13, line 469. The sentence is not complete, which should be fixed. 

Author Response

Reviewer 2 was generally positive and wrote that “experiment is well designed and these authors considered many aspects of the effect and performed corresponding measurements. (…) I believe the public as well as the academics need to know all the experiments and results asap”
Three remarks require our attention:

  1. It is no doubt that Omicron should be closed to test the TMPRSS2's effect. However, I cannot find why these authors chose the Beta variant, especially at the beginning.  There is indeed one sentence for this part at the beginning of the manuscript.  "While the early SARS-CoV-2 variants could only grow in humanized mice expressing the human ACE2 receptor, the Beta variant (B.1.351) and the Omicron variant (B.1.1.529) can infect C57BL/6 mice and trangenic mice on this background (including TMPRSS2-/- mice) due to naturally occurring S protein mutation N501Y within the receptor binding domain (RBD) ." However, alpha and gamma variants contain the N501Y mutation on their RBD (1-2). So, the authors should elaborate on the reasoning. 
  2. N501Y mutation of spike protein in sars-cov-2 strengthens its binding to receptor ace2. Elife 2021, 10, e69091.
  3. The N501Y spike substitution enhances sars-cov-2 infection and transmission. Nature 2022, 602, 294-299.

Thank you for this important question. We opted for the Beta variant partly for historical reasons, as it was readily available in our lab, and in part because it was shown that beta elicits more severe disease in mice than the alpha or the gamma variant (Montagutelli et al. BiorXiv 2021). Therefore, we opted for one representative pre-omicron variant to demonstrate the relevance of TMPRSS2 in its in vivo replication, and to contrast it to Omicron SARS-CoV-2. While other variants such as Alpha or Gamma could have been added as well, we decided that such datasets would add only a marginal benefit to the manuscript. We amended the sentence in lines 67-71 to make clear that Beta and Omicron are not the only viruses that could have been used. Also, we added a sentence in lines 252-253 to explain the rationale for choosing beta over alpha or gamma.

  1. In this work, the authors examined the effect of TMPRSS2 using mouse model. Although many effect of the protease for Omicron is still observed in mouse, the effect on humans may be slightly different. This point should be kept in mind.    

We agree and have in mind that additional factors may play a role in human infection with SARS-CoV-2. Therefore, we have discussed this limitation of our study in the discussion (lines 476-478).

  1. Page 13, line 469. The sentence is not complete, which should be fixed. 

Thank you for bringing this to our attention. We have now fixed the incomplete sentence.

Reviewer 3 Report

  • Major comments: 

In the article, Kristin Metzdorf et al. investigated the contributions of TMPRSS2 to the spread of SARS-CoV-2 Beta and Omicron variants in C57BL/6-TMPRSS2-/- and C57BL/6-TMPRSS2WT/WT mice models, and found that the loss of TMPRSS2 strongly reduced the replication of the Beta variant in nose, trachea, and lung of C57BL mice and protected the animals from weight loss and disease, while the infection of mice with the Omicron variant did not cause disease, but TMPRSS2 was essential for efficient viral spread in the upper and lower respiratory tract.

The study is relevant to the field and well-organized.

Here are some considerations and suggestions for the study:

  • Specific comments:

1)      Line 65-70, why the study chose the Beta variant over the alpha, and Gamma variants, as they all bear N501Y mutation within the receptor binding domain (RBD)?

2)      Line 109, please provide proper citations here.

3)      Why were the body mass and clinical conditions of the mice only monitored for three days?

4)      Line 140, please give the full name of “CBV” here.

5)      Line 151-158, please provide the rationale for why the study chooses RSP9 as the reference gene over GAPDH.

6)      Line 151-158, the IFNs analyses were only based on the gene level. However, all these IFNs will exert their immune response only at the protein level, which is not directly related to their gene expression levels, and thus, there must be a comparison of these markers at the protein level, too, to make a clear conclusion.

7)      Line 198, please provide the rationale for why the study chooses MAC2 to label macrophages.

8)      Supplementary Figure 1, How many animals were used for the young and aged animals, respectively? As this information is not available in the methods section (especially young mice). Besides, the data was shown here as mean ± SEM?

9)      Line 300-302, and Supplementary Figure 2, is it possible that 3 days of SARS-CoV-2 infection in mice is too short to be detected in brains? Besides, Supplementary Figure 2B, a typo of “Life-virus”.

10)   Line 302-302, evidence showed that Omicron could replicate in the upper and lower respiratory tracts in the C57BL/6 mouse model. It is weird here that no infectious Omicron virus was detected at all in Fig. 2D as the SARS-CoV-2 N gene could be detected in Fig. 2B.

11)   Line 325-333 and Fig. 3B, 3C, no statistical analysis in Fig. 3B and 3C? Besides, please specify how many cells were counted and how many were positive in Fig. 3B, C?

12)   Line 357-358, the description mentioned here “while infiltration measured for wt and TMPRSS2 KO mice were within the background range”, refers to the infection with the SARS-CoV-2 Omicron variant?

13)   What do "M", "B", "I", "V", and "P" mean in Figure 4A?

14)   Line 367, typo of “attaining”.

15)   Line 374, “1” = occasionally seen.

16)   Line 469, showing what?

17)   Line 499, typo of “1x104”.

18)   Line 500, typo of “+”.

Author Response

Reviewer 3 praised our study as “relevant to the field and well-organized” and had several suggestion on how to improve our manuscript:

  • Specific comments:

1) Line 65-70, why the study chose the Beta variant over the alpha, and Gamma variants, as they all bear N501Y mutation within the receptor binding domain (RBD)?

As stated in the answer to Reviewer 2, the Beta variant induced more severe pathology than alpha or gamma and was readily available in our lab. However, it is likely that any other N501Y variant would have been OK as well. We amended the text in lines 67-71 accordingly and added in lines 252-253 a brief explanation for the rationale of using the beta over the alpha or gamma variants. 

2)      Line 109, please provide proper citations here.

Thank you for pointing this out, we have now updated it.

3)      Why were the body mass and clinical conditions of the mice only monitored for three days?

The duration of the infection of the mice in this experiment was limited to 3 days before animals were sacrificed for organ harvest, which allowed us to use the same animals for two readouts, in line with the 3R guidelines of animal welfare. Therefore, the body mass and scores have been observed and evaluated only for a brief period, yet we could identify clear TMPRSS2 related effects in Beta-infected mice. Others have published, that BA.1-infection does not cause clinical disease in mice at later time-points, which fits our unpublished data. Hence, we saw no purpose in additional monitoring of mice over the initial 3 days. We consider this a minor point.

4)      Line 140, please give the full name of “CBV” here.

The “lysis solution CBV” is a component of the RNA isolation kit from innuPrep Virus TS RNA kit (Analytik Jena) and the only name used by the manufacturer. Since the kit instructions do not list the full name, we have now set it to italics to indicate that CBV is a name and not an acronym.

5)      Line 151-158, please provide the rationale for why the study chooses RSP9 as the reference gene over GAPDH.

The murine ribosomal protein 9 (Rsp9) has proven to be a reliable housekeeping reference gene for cytokine determination by qPCR, and is regularly used by the Bruder lab, which performed these readouts within the study. While GAPDH would likewise be a possible candidate, the use of Rsp9 for the normalization of cytokines using the ΔCT method has already been well-established and we opted for the available method, rather than for changes in the protocol.

6)      Line 151-158, the IFNs analyses were only based on the gene level. However, all these IFNs will exert their immune response only at the protein level, which is not directly related to their gene expression levels, and thus, there must be a comparison of these markers at the protein level, too, to make a clear conclusion.

While we agree with the reviewer that protein levels may be misleading in certain circumstances, we disagree that they are not directly related to the gene expression of interferons. Interferons are typically expressed in response to immune sensing upon IRF3 or IRF7 elicited de novo transcriptional activity, rather than being sequestered in granules and released upon activation. Nevertheless, we agree that protein measurements would have been a more accurate and direct measure of IFN responses. Since we do not have any more sample materials to perform such measurements, and since the IFNs and their inflammatory effects play a minor role in this manuscript, we have decided to place the figure in the supplementary materials. Moreover, we added a statement to caution that IFN was measured on the mRNA and not on the protein level (lines 363-364).

7)      Line 198, please provide the rationale for why the study chooses MAC2 to label macrophages.

MAC2 and F4/80 are the markers of choice for immunohistological analysis in our histology unit, but MAC2-specific macrophages are more stable after lung development, which allowed us to determine their numbers in both young and old mice. Hence, this marker was used as a trustworthy staining of macrophages in the lungs, regardless of the animal’s age.

8)      Supplementary Figure 1, How many animals were used for the young and aged animals, respectively? As this information is not available in the methods section (especially young mice). Besides, the data was shown here as mean ± SEM?

Thank you for this important comment. We have amended the supplementary figure 1 legend to add the specified information (n=8 / group in young mice, n=8-9 / group in old mice). Each experiment was performed independently at least twice and the results were pooled.

9)      Line 300-302, and Supplementary Figure 2, is it possible that 3 days of SARS-CoV-2 infection in mice is too short to be detected in brains? Besides, Supplementary Figure 2B, a typo of “Life-virus”.

Published literature agrees that SARS-CoV-2 in brains of hACE2 mice can be identified as early as 2-3 days post infection (Winkler et al. Nat. Immunol. 2020, Kumari et al. Viruses 2021). Hence, we are certain that the virus was undetectable in brains of WT mice at a time when it was present in hACE2, according to literature. While we cannot formally exclude that the virus might become detectable in the brain of our mice at a later time point, Beta-infected mice would have reached the humane endpoint on day 4 (figure 1 and supplementary figure 1), which derails efforts to analyze them at substantially later time-points. We added a statement to the discussion to indicate that our analysis was restricted to dpi 3 (lines 446-447).

The Spelling mistake has been corrected, thank you.

10)   Line 302-302, evidence showed that Omicron could replicate in the upper and lower respiratory tracts in the C57BL/6 mouse model. It is weird here that no infectious Omicron virus was detected at all in Fig. 2D as the SARS-CoV-2 N gene could be detected in Fig. 2B.

It is not very surprising that omicron virus could only be detected with the more sensitive method. This has already been pointed out in the manuscript, and we consider it a minor point.

11)   Line 325-333 and Fig. 3B, 3C, no statistical analysis in Fig. 3B and 3C? Besides, please specify how many cells were counted and how many were positive in Fig. 3B, C?

Several thousand cells were counted in the tissue, both all cells and the cells stained with SARS-CoV-2 and MAC2. The number of cells counted varied between the individual lung sections of each mouse. For these reasons, we report the positive cells as a percentage.
We have performed statistical analysis on these results, but due to their complexity, we decided that the results are underpowered for conclusions drawn out of statistics. Namely, t-test comparisons of individual pairs of columns typically resulted in statistically significant differences between infected WT mice and controls, but ANOVA analysis resulted in no statistically significant differences, likely due to penalization for multiple comparisons. In consequence, we cannot define if the lack of statistical significance in our ANOVA studies is due to a bona find lack of significance or to a lack of statistical power due to the size of groups that were used (typically, n=8-10 mice per group).
To avoid misleading interpretations, we opted for descriptive, rather than analytical statistics in this figure.

12)   Line 357-358, the description mentioned here “while infiltration measured for wt and TMPRSS2 KO mice were within the background range”, refers to the infection with the SARS-CoV-2 Omicron variant?

Yes, we clarified this now in line 358.

13)   What do "M", "B", "I", "V", and "P" mean in Figure 4A?

These Letters refer to the individual parameters that represent the lung inflammation score mentioned in the Material and Method Section “Histology”.

We understand that this was not clear in the respective figure legend and adapted this explanation in the manuscript Figure 4A as follows:

“(A) Representative hematoxylin-eosin (HE) staining of lung tissue from PBS (top), Beta (middle) and Omicron (bottom) infected wild-type (left) and TMPRSS2 KO mice (right).I= interstitial pneumonia, B= broncho-alveolar inflammation, P= perivascular inflammation and V= vasculitis, M= macrophages representing the severity of the individual parameters summarized in the Pneumonia score. Scale bar 50 µm”.

Reviewer comments 14-18:

Thank you for bringing the following typing mistakes to our attention. We have now adapted the correct spellings in the manuscript.

14)   Line 367, typo of “attaining”.

15)   Line 374, “1” = occasionally seen.

16)   Line 469, showing what?

17)   Line 499, typo of “1x104”.

18)   Line 500, typo of “+”.

Round 2

Reviewer 3 Report

I believe that manuscript has been sufficiently improved, and the authors have sufficiently addressed most of my concerns. 
